# Bundle Networks: Fiber Bundles, Local Trivializations, and a Generative Approach to Exploring Many-to-one Maps

**Nico Courts** [*]
University of Washington
`ncourts@math.washington.edu`

**Henry Kvinge**
Pacific Northwest National Laboratory,
University of Washington
`henry.kvinge@pnnl.gov`

## Abstract

Many-to-one maps are ubiquitous in machine learning, from the image recognition model that assigns a multitude of distinct images to the concept of "cat" to the time series forecasting model which assigns a range of distinct time-series to a single scalar regression value. While the primary use of such models is naturally to associate correct output to each input, in many problems it is also useful to be able to explore, understand, and sample from a model's fibers, which are the set of input values $x$ such that $f(x) = y$, for fixed $y$ in the output space. In this paper we show that popular generative architectures are ill-suited to such tasks. Motivated by this we introduce a novel generative architecture, a Bundle Network, based on the concept of a fiber bundle from (differential) topology. BundleNets exploit the idea of a local trivialization wherein a space can be locally decomposed into a product space that cleanly encodes the many-to-one nature of the map. By enforcing this decomposition in BundleNets and by utilizing state-of-the-art invertible components, investigating a network's fibers becomes natural.

## 1 Introduction

In the last decade, generative methods have made remarkable strides toward the goal of faithfully modeling complex data distributions. Families of model architectures such as generative adversarial networks (GANs) (Goodfellow et al., 2014), variational autoencoders (VAEs) (Kingma & Welling, 2013), and normalizing flows (Rezende & Mohamed, 2015) have enabled a range of tasks that were previously thought to be unachievable for complex (and often high-dimensional) distributions such as those representing natural images. Applications include image to image translation (Mao et al., 2019; Lee et al., 2018), music generation (Mogren, 2016), image super-resolution (Ledig et al., 2017), and density estimation (Dinh et al., 2016). In this work we introduce a new task in machine learning which revolves around the problem of modeling the structure of many-to-one maps.

The majority of classification and regression tasks in machine learning involve many-to-one maps. That is, maps that send a range of distinct inputs to the same output. For example, most image recognition tasks are many-to-one since they send an infinite set of natural images to a finite number of classes. Of course, there is no reason that a model's target space needs to be discrete—consider, for instance, a sentiment analysis task where sentiment is naturally represented by a continuum. There are likely a range of distinct expressions that result in the same sentiment value. This paper was inspired by a problem in materials science wherein one wishes to explore all the processing conditions that will result in a material with a specific fixed property (e.g., strength). Understanding all such processing conditions allows the materials scientist to choose among a range of options that will all produce a material achieving their desired performance metrics.

While the focus of most research on classification and regression tasks is training models that give correct predictions, it can also be useful to be able to fix an output value $y$ and explore, quantify, or sample from all those $x$ which map to $y$. Such an exercise allows one to understand different modes of variation that a model collapses in the process of making its predictions. If function $\pi : X \to Y$

---

[*]Work completed during an internship at Pacific Northwest National Laboratory.

is the ground truth pairing between input space $X$ and output space $Y$, this amounts to calculating the inverse image $\pi^{-1}(y)$, or *fiber* of $\pi$ at $y$.

Our goal in the present work is to (1) formalize this problem and (2) describe a deep learning framework which readily enables this kind of analysis. We take as our inspiration the notion of a fiber bundle from topology (Seifert, 1933; Whitney, 1935). Consider the projection map on $X = Y \times Z$, $\pi : Y \times Z \to Y$, which sends $\pi(y, z) = y$. For any $y \in Y$, the inverse image $\pi^{-1}(y)$ is easily calculated as $\{y\} \times Z \cong Z$. Interpreted in terms of the machine learning task, $Y$ is the component of $X$ that we want to predict and $Z$ encodes the remaining variation occurring among all $x$ that map to $y$. Unfortunately, in our nonlinear world data distributions can rarely be decomposed into a product space on a global level like this. Instead, it is more realistic to hope that for each sufficiently small neighborhood $U$ of $Y$, we can find a data distribution preserving homeomorphism $U \times Z \xrightarrow{\sim} \pi^{-1}(U)$. This informally summarizes the idea of a fiber bundle, where locally the space $X$ is the product of a base space (some subset of $Y$) and a fiber $Z$. Following the mathematical literature, we call this task *local trivialization of $X$*. A local trivialization is useful since it allows us to understand and explore the fibers of $\pi$.

In our experiments we show that popular generative models including Wasserstein GANs (Wu et al., 2018) and conditional GANs (Mirza & Osindero, 2014), are unable to accurately model the fiber structure of a machine learning task. To address this, we describe a new deep learning architecture called a *Bundle Network* which is designed to produce local trivializations of an input space $X$ and thus enable one to model fibers of a task. Bundle Networks uses a clustering algorithm to divide up the output space $Y$ into neighborhoods $\{U_i\}$ and then learn a conditioned, invertible mapping that decomposes the inverse image of each of these neighborhoods $\pi^{-1}(U_i)$ into a product $U_i \times Z$ that preserves the distributions induced by the data. We show that Bundle Networks can effectively model fibers for a range of datasets and tasks, from synthetic datasets that represent classic examples of fiber bundles, to the Wine Quality (Cortez et al., 2009) and Airfoil Noise (Brooks et al., 1989) datasets from (Dua & Graff, 2017).

In summary, our contributions in this paper include the following.

- We identify and formalize the problem of *learning the fibers* of a machine learning task.

- We describe one approach to solving this problem, using a fiber bundle framework to learn how to locally decompose the input space of the task into a product of label space and a fiber space that parametrizes all additional variation in the data.

- We introduce a new family of deep generative models called Bundle Networks and show via empirical experiments that, while other popular architectures struggle to model the fibers of a machine learning task, Bundle Networks yield strong results.

## 2 RELATED WORK

**Conditional and manifold-based generative models:** The local trivialization task that we describe in Section 3.1 can be interpreted as a special type of conditional generative task. That is, a task where data generation is conditioned on some additional variables. Common models within this framework are the various flavors of conditional variational autoencoders (VAE) (Sohn et al., 2015) and conditional generative adversarial networks (GAN) (Mirza & Osindero, 2014). While our approach utilizes conditional components to capture differences in model behavior at different locations in input space, additional aspects of the task, such as the local decomposition of input space into fiber and label space, mean that standard generative models do not give strong results.

This paper also shares similarities to other work at the intersection of generative modeling and manifold theory. In Kalatzis et al. (2021), the authors propose multi-chart flows (MCFs) for learning distributions on topologically nontrivial manifolds. Similar to the present work, the authors use coordinate charts to learn non-Euclidean distributions using otherwise Euclidean machinery. Unlike our work, MCFs are designed to learn general distributions on manifolds and do not have the specialized architectural adaptations or novel training routine designed for modeling many-to-one problems. Vector bundles have also been applied to data science problems in Scoccola & Perea (2021); Ye & Lim (2017); Gao et al. (2021); Knöppel & Pinkall (2016).

**Invertible neural networks:** Invertible neural networks have recently become a topic of interest within the deep learning community. Some of the first network layers that were both invertible and learnable were suggested in (Dinh et al., 2015; 2016) and are in essence affine transformations followed by orthogonal transformations. Ardizzone et al. (2018) and Sorrenson et al. (2020) are among the works that have used invertible networks to solve problems of distribution approximation with the latter introducing general incompressible-flow networks (GINs) which are volume-preserving versions of invertible neural networks. The most common instance of invertible neural networks are normalizing flows, which were popularized by (Rezende & Mohamed, 2015).

**Feature disentanglement:** Distribution disentanglement is (roughly) the task of learning a latent space for a distribution such that orthogonal directions capture independent and uncorrelated modes of variation in the data. Many recent methods of solving this problem leverage popular generative models such as generative adversarial networks (GANs) and variational autoencoders (VAEs) (Chen et al., 2018; Kim & Mnih, 2018; Dupont, 2018; Chen et al., 2016; Lin et al., 2020). Like disentanglement, the local trivialization task described in this paper involves factoring a data distribution. Unlike the problem of feature disentanglement, which generally seeks to decompose a distribution into all independent modes of variation, we only seek to distinguish between those modes of variation defined by the classes or targets of the supervised task and all other variation. The local nature of our factorization further sets us apart from work on feature disentanglement.

## 3 FIBER BUNDLES AND LOCAL TRIVIALIZATIONS

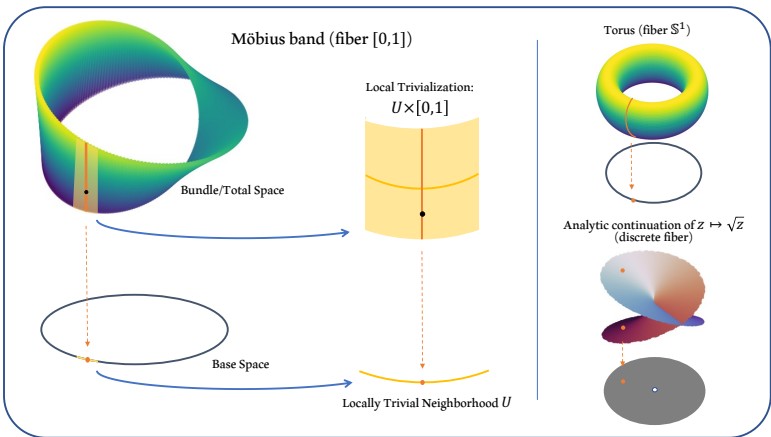

Figure 1: Examples of fiber bundles. (**Left**) A Möbius band as a bundle over the circle $\mathbb{S}^1$. A typical fiber over a base point (dark orange) and a locally trivial neighborhood and local trivialization are included for illustration. (**Right**) Two other examples of fiber bundles. Above is the Torus (with circular fiber) lying over the circle. Below is the analytic continuation of the complex function $z \mapsto \sqrt{z}$ over the punctured disc $\widehat{\mathbb{D}} = \{z \in \mathbb{C} | 0 < |z| < 1\}$, which has a two-point fiber.

One way to understand complicated geometric spaces is to decompose them into simpler components. For example, topologically a torus $T$ can be understood to be the product of two circles $T \cong S^1 \times S^1$. Such decompositions into product spaces are particularly useful in applications since many algorithms on the entire space can then be reduced to independent, component-wise calculations. However, even in situations where local neighborhoods look like a product, such decompositions may not be possible globally because of "twists" that occur in a space (for example in the Möbius band, see Figure 1). First developed in the work of Seifert (1933) and Whitney (1935), fiber bundles are a geometric structure designed to capture the **locally** product-like nature of such spaces.

Formally, a *fiber bundle* is a triple $(E, B, Z)$ of topological spaces along with a (surjective) continuous map $\pi : E \rightarrow B$ and a collection of *local trivializations* $\mathcal{T} = \{(U_i, \varphi_i)\}_{i \in \mathcal{I}}$ subject to the conditions that $(i)$ $\{U_i\}_{i \in \mathcal{I}}$ is an open cover of $B$ and $(ii)$ for each $i \in$

$\mathcal{I}$, $\varphi_i$ gives an homeomorphism $\varphi_i : U_i \times Z \xrightarrow{\cong} \pi^{-1}(U_i)$ such that the obvious projection from $U_i \times Z$ to $U_i$ agrees with the map $\pi$ restricted from $\pi^{-1}(U_i)$ to $U_i$. This last condition is equivalent to saying that for all $i \in \mathcal{I}$, the following diagram commutes. The space $E$ is generally called the *total space*, $B$ is called the *base space*, and $Z$ is called the *fiber*. One can think of this definition as rigorously capturing the idea that for small enough neighborhoods, $E$ looks like the product of a subset of $B$ and $Z$. Some standard fiber bundles are found in figure 1.

$$\begin{array}{ccc} \pi^{-1}(U_i) & \underset{\varphi_i^{-1}}{\overset{\varphi_i}{\rightleftarrows}} & U_i \times Z \\ \pi \downarrow & \swarrow \text{proj} & \\ U_i & & \end{array} \tag{1}$$

### 3.1 THE LOCAL TRIVIALIZATION OF A MACHINE LEARNING TASK AND MODELING DATA ON A FIBER

Suppose that $\mathcal{T}$ is a supervised machine learning task that involves taking data from $X$ and predicting a correspond element of $Y$. Let $\mathcal{D}_X$ be the data distribution on $X$ and let measurable function $\pi : \mathrm{supp}(\mathcal{D}_X) \to Y$, define a ground truth label for each $x$ in the support of $\mathcal{D}_X$. Then for any measurable open set $U \subseteq Y$, $\mathcal{D}_X$ induces a conditional distribution, $\mathcal{D}_{\pi^{-1}(U)}$ on $\pi^{-1}(U)$. In a similar fashion, via the pushforward of $\mathcal{D}_X$ using $\pi$, we get a label distribution $\mathcal{D}_Y$ on $Y$. Finally, each measurable subset $U \subseteq Y$ inherits a conditional distribution $\mathcal{D}_U$ from $\mathcal{D}_Y$. We call the sets $\pi^{-1}(U)$ with distribution $\mathcal{D}_{\pi^{-1}(U)}$ the *trivializable neighborhoods of task $\mathcal{T}$*.

A *local trivialization* of $X$ with respect to $\pi$, involves:

1. A choice of open cover $\mathcal{U} = \{U_i\}$ of $Y$ such that $\cup_i U_i = Y$.
2. A choice of fiber space $Z$ with associated *fiber distribution $\mathcal{D}_Z$*.
3. Learning an invertible, *trivializing map* $\varphi_i : U_i \times Z \to \pi^{-1}(U_i)$ for each $U_i \in \mathcal{U}$, such that (a) $\varphi_i$ maps distribution $\mathcal{D}_{U_i} \otimes \mathcal{D}_Z$ to $\mathcal{D}_{\pi^{-1}(U_i)}$ and (b) $\varphi_i^{-1}$ makes Diagram 1 commute.

Note that in the above formulation, the space $X$ takes the role of the total space $E$ from section 3 and $Y$ takes the role of the base space $B$. Though good choices of fiber $Z$ and measure $\nu$ can occasionally be guessed based on the nature of the data, they generally must be arrived at through experimentation. Further, the use of local trivializations should be seen as a convenient framework that enables the input space to be locally approximated as the product of two subspaces: one corresponding to the label and one capturing variation unrelated to the label. One should not expect that a task and its fibers are literally a fiber bundle "on the nose."

The trivializing maps $\{\varphi_i\}_i$ allow one to explore the fibers of $\pi$, which includes sampling from them. For example, if we choose some $y \in Y$ then we can sample from the fiber $\pi^{-1}(y)$ by (1) identifying the neighborhood $U_i$ that $y$ belongs to, (2) sampling a collection of points $\{z_1, \ldots, z_\ell\}$ using $\mathcal{D}_Z$, and (3) applying the map $\varphi_i$ to $(y, z_i)$ for each $z \in \{z_1, \ldots, z_\ell\}$.

## 4 BUNDLE NETWORKS

We introduce a new model called *Bundle Network* (or *BundleNet* for short) to address the problem described in section 3.1. A BundleNet (sketched out in figure 2) combines an invertible architecture with the ability to condition on elements of $\mathcal{R}$ which are in bijective correspondence with the neighborhoods $\mathcal{U}$ of $Y$. Specifically, BundleNet is a function $\Phi : X \times \mathcal{R} \to Y \times Z$, such that given a specific choice of conditioning vector $r \in \mathcal{R}$, $\Phi_i := \Phi(-, r_i) : X \to Y \times Z$ is an invertible neural network. $\Phi_i$ and $\Phi_i^{-1}$ are meant to model $\varphi_i^{-1}$ and $\varphi_i$ (respectively). Moving from forward to back, the first layer of $\Phi$ is an affine transformation conditioned (solely) on elements from $\mathcal{R}$. After this layer, the model consists of alternating RNVP blocks and coordinate permutations.

As noted in section 3.1, a number of choices need to be made before a model is trained. Let $D_t \subseteq X \times Y$ be the training set and $p_X : X \times Y \to X$ and $p_Y : X \times Y \to Y$ be the standard projections of $X \times Y$ onto its components. Firstly, the collection of neighborhoods $\mathcal{U}$ in $Y$ needs to be chosen. In the case of regression, where $p_Y(D_t)$ can contain an arbitrary number of distinct elements, we found that using $k$-means was an efficient and effective way to cluster these points into neighborhoods. The set of cluster centroids are then exactly the vectors $\mathcal{R} = \{r_1, \ldots, r_q\}$ that will be used to condition

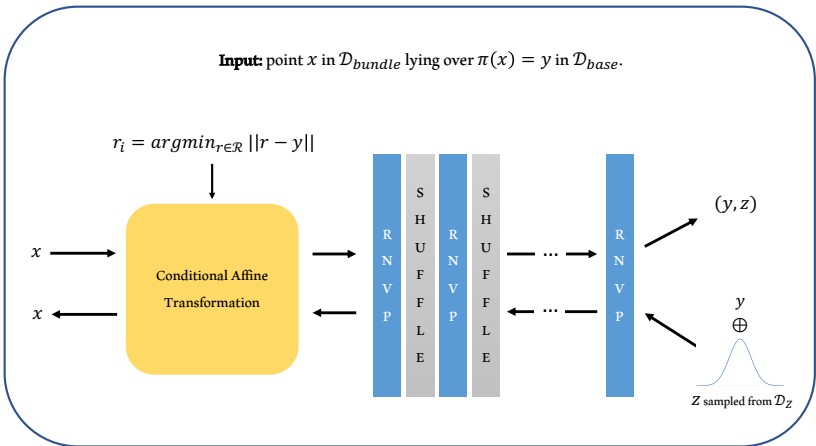

Figure 2: A diagram of the BundleNet architecture. The upper arrows indicate the forward operation of the model while the lower arrows indicate the reverse operation of the model.

the network. The number of clusters, $q$, is a hyperparameter that can be explored during training (we discuss the robustness of BundleNet to different choices of $q$ in section A.3.1). In the case where the output values are finite (i.e. $p_Y(D_t)$ can only take a finite number of values $C = \{c_1, \ldots, c_q\}$), one can simply set $U_i = \{c_i\}$. After selecting neighborhoods $\mathcal{U}$ in the base space, the neighborhoods in $X$ are defined automatically by taking an inverse image with respect to $\pi$. The fiber $Z$ and fiber distribution $\mathcal{D}_z$ also need to be chosen. These choices are discussed in the first part of Section 6.

We call application of the model in the direction $X \to Y \times Z$ *forward operation* and application of the model in the direction $Y \times Z \to X$ *reverse operation* in accordance with the ground truth association function $\pi : X \to Y$. As depicted in Figure 2, to apply $\Phi$ to a training point $(x, y) \in D_t$ in the "forward direction" we solve $r_i = \arg\min_{r \in \mathcal{R}} ||r - y||$, finding the representative $r_i$ in $\mathcal{R}$ that is closest to label $y$. Then we apply $\Phi_i = \Phi(-, r_i)$ to $x$. To run $\Phi$ in reverse on training example $(x, y)$ we invert $\Phi_i$ to $\Phi_i^{-1}$, sample some $z$ from $\mathcal{D}_Z$, and compute $\Phi_i^{-1}(y, z)$. To ease notation we write $\Phi(x)$ (respectively, $\Phi^{-1}(y, z)$) to denote the process of first computing $r_i$ and then applying $\Phi_i$ to $x$ (resp. $\Phi_i^{-1}$ to $(y, z)$).

Since we are primarily interested in generating and sampling from task fibers via their local trivialization, at inference time we only run $\Phi$ in reverse, applying $\Phi_i^{-1}$ to a collection of points $(y, z)$ sampled from $\mathcal{D}_{U_i} \otimes \mathcal{D}_Z$. We evaluate this mapping by measuring the discrepancy of the resulting image in $X$ against the distribution $\mathcal{D}_{\pi^{-1}(U_i)}$. One could conceivably also use $\Phi$ for classification, but in that case one would need to develop a strategy for identifying the proper $r_i \in \mathcal{R}$ to use to condition $\Phi$ in the absence of a label associated with $x \in X$. One could likely use a nearest neighbor approach using points that are known to belong to each $\pi^{-1}(U_i)$, but we leave this for future work.

The loss we choose for our model consists of two main parts, $\mathcal{L}_{\text{fwd}}$ and $\mathcal{L}_{\text{bwd}}$, corresponding to loss from the forward and backward directions. Since $\Phi$ is invertible, these can be trained simultaneously and updated with a unified gradient step. Together, our final loss is then $\mathcal{L}_{\text{bundlenet}} = \mathcal{L}_{\text{bwd}} + \mathcal{L}_{\text{fwd}}$.

The forward loss $\mathcal{L}_{\text{fwd}}$ is the simpler of the pair. Given $(x, y) \in D_t$, we find $i$ such that $y \in U_i$ and compute $\mathcal{L}_{\text{fwd}}(x, y) = ||p_Y(\Phi_i(x)) - y||^2$, where $p_Y$ denotes the projection from $Y \times Z$ to $Y$. This term ensures that in the forward direction $\Phi$ accurately maps elements from $X$ to base space $Y$. The backward loss is comprised of several terms: $\mathcal{L}_{\text{bwd}} = \mathcal{L}_{\text{KL−fwd}} + \mathcal{L}_{\text{KL−bwd}} + \mathcal{L}_{\text{MSMD}}$ (where MSMD is mean squared minimum distance). Choosing some $U_i \in \mathcal{U}$, we sample points from $\mathcal{D}_{U_i}$ and pair each with a point that we sample from our priors (that is, from $\mathcal{D}_z$) to get a set $S$. $S$ is passed through $\Phi_i^{-1}$ to generate points in $X$. Then the MSMD, KL-fwd, and KL-bwd losses (cf. section A.2) are used to compute a loss term by comparing $\Phi_i^{-1}(S)$ and points sampled from $\mathcal{D}_{\pi^{-1}(U_i)}$. The KL-divergence terms act to ensure that the point clouds $\Phi_i^{-1}(S)$ and $\pi^{-1}(U_i)$ cover one another and the MSMD term ensures that no point in $\Phi_i^{-1}(S)$ is too far from some point in $\pi^{-1}(U_i)$.

# 5 EXPERIMENTS

We evaluate Bundle Networks on four datasets. Two of them are synthetic and represent classic examples of fiber bundles: the Möbius band and the torus, both embedded in $\mathbb{R}^3$. The other datasets are Wine Quality (Cortez et al., 2009) and Airfoil Noise (Brooks et al., 1989), both of which are available in the UC Irvine Machine Learning Repository (Dua & Graff, 2017). The Wine Quality task (as we have structured it) involves predicting wine quality and color (white vs. red) from 11 measured quantities. The Airfoil Noise task has 5 real-valued input parameters and one predicted output (scaled sound pressure level). Further information can be found in section A.1. For the invertible layers in BundleNet we used the `FrEIA` package (Ardizzone et al., 2021). We chose Wine Quality and Airfoil Noise because they represent low-dimensional, many-to-one relationships with several continuous variables making each a good first test case for the local trivialization task.

In selecting models against which to compare our BundleNets, we chose modern yet general-purpose models that have proven effective at many generative tasks. This includes Wasserstein divergence GANs (WGAN) developed as an improvement on traditional Wasserstein GANs by Wu et al. (2018) and conditional GANs (CGANs) from Mirza & Osindero (2014). While WGAN directly learns to approximate the global distribution of data using Wasserstein distance, it doesn't include a conditional aspect that allows a user to specify a fiber to sample from so it is excluded from our fiberwise metrics. We also implement a version of a CGAN that we call CGAN-local that conditions on a concatenated vector consisting both of the base point and a neighborhood representative, chosen in a manner analogous to BundleNet. We developed this model to identify whether a CGAN could perform on par or better than BundleNet when it is provided with the exact same locality information during training and testing. As we describe in section 5.2, we take CGAN-local's failure to reach BundleNet's level of performance as evidence that the specialized architecture and loss functions BundleNet uses makes it better equipped for the task of fiber modeling.

## 5.1 TRAINING AND EVALUATION PROCEDURE

Each model was trained on a single GPU (Nvidia Tesla p100 @ 16GB) for a period of 2000 epochs using an initial learning rate of $10^{-4}$ which was halved every 70 epochs. The weights that were used for evaluation were the final weights that were saved after the final epoch, even if the model converged long before. A table of hyperparameters used for these results (including fiber distributions $\mathcal{D}_Z$) can be found in Appendix section A.3 along with a discussion of their impact on performance.

The aim of our evaluation is to understand how well BundleNet models can reconstruct fibers in the total space $X$. We use a number of different metrics to do this including: MSMD, KL-divergence, Wasserstein metrics ($\mathcal{W}_1$ and $\mathcal{W}_2$), and maximum mean discrepancy (MMD). We perform such measurements in two modes: the *global* and *fiberwise* regimes. In the global regime, the trained model is asked to generate 4000 points from the global distribution $D_X$ (using points sampled from $Y$ and $Z$) and these are compared against 4000 points from either the true data distribution (in the synthetic datasets when this is known) or from a held-out test set (when we don't have access to the underlying distribution). Then these two point clouds are evaluated using each of our metrics. The process is repeated 10 times and the output values are bootstrapped to compute sample mean with 95% confidence intervals. In the fiberwise regime, 15 points are sampled from $Y$ and for each such $y$, we use the model to generate 200 points lying in $\pi^{-1}(y)$. We compare this against 200 points we have sampled from either the true fiber (when this is known) or the points lying over the base point in our combined train & test sets. Each metric is evaluated 10 times and bootstrapped as above.

## 5.2 RESULTS

For the results described in this section we mostly focus on the Wasserstein-1 metric $\mathcal{W}_1$ as our evaluation statistic. The full suite of results (MSMD, MMD, KL-fwd, KL-bwd, $\mathcal{W}_1$, and $\mathcal{W}_2$) can be found in Appendix section A.5. With a few exceptions, performance patterns we see for the Wasserstein-1 metric are largely repeated when we apply other metrics.

As can be seen in tables 1 and 2, with the exception of CGAN-local's performance on Airfoil Noise in the fiberwise regime, BundleNet outperforms all the other models, sometimes by a very wide margin, in both the global and fiberwise data generation regimes. In fact, the only dataset where other models are competitive with BundleNet is Airfoil Noise and that is again only in the fiberwise

Table 1: The Wasserstein-1 metric for global data generation across all four datasets and models. Each metric is applied to a trained model as detailed in section 5.1 with 95% confidence intervals.

|  | BundleNet (ours) | WGAN | CGAN | CGAN-local |
|---|---|---|---|---|
| Torus | **0.461 ± 0.020** | 2.191 ± 0.011 | 7.114 ± 0.015 | 1.610 ± 0.031 |
| Möbius band | **0.264 ± 0.004** | 2.048 ± 0.013 | 1.844 ± 0.015 | 6.497 ± 0.101 |
| Wine Quality | **1.733 ± 0.011** | 99.22 ± 0.392 | 4.054 ± 0.014 | 2.114 ± 0.011 |
| Airfoil Noise | **1.448 ± 0.022** | 4.741 ± 0.034 | 2.508 ± 0.025 | 1.735 ± 0.013 |

Table 2: The Wasserstein-1 metric for fiberwise data generation across all four datasets and the three models that have a conditional component. Intervals represent 95% CIs over 10 trials.

|  | BundleNet (ours) | CGAN | CGAN-local |
|---|---|---|---|
| Torus | **0.251 ± 0.013** | 7.996 ± 0.015 | 0.450 ± 0.018 |
| Möbius band | **0.279 ± 0.011** | 9.860 ± 0.038 | 8.228 ± 0.904 |
| Wine Quality | **1.917 ± 0.172** | 3.666 ± 0.233 | 2.926 ± 0.169 |
| Airfoil Noise | 3.124 ± 0.089 | 3.563 ± 0.158 | **3.076 ± 0.063** |

regime (table 10, bottom). Here BundleNet achieves better performance than CGAN and CGAN-local on 4 out of 6 metrics (though on 2 of these BundleNet's 95% confidence interval intersects another model's). Overall, CGAN-local is the model that is most competitive with BundleNet.

Within the global data generation regime, BundleNet easily outperforms the other three models on all four datasets. CGAN-local comes in second on three datasets (Torus, Wine Quality, and Airfoil Noise) while CGAN comes in second on one (Möbius band). We were surprised to find that though BundleNet is specifically designed for the purpose of sampling from fibers, it also achieves high accuracy when reconstructing the whole distribution. Visualizations of each model's reconstruction of the Torus and Möbius band can be found in figure 6 in the appendix with further commentary.

As remarked above, we developed CGAN-local so that we would have a GAN-based benchmark that receives the exact same input as BundleNet. We hoped to be able to ascertain whether or not BundleNet's specialized invertible architecture and loss function combination is critical to better model performance. The results above suggest that BundleNet is indeed better adapted to solving the problem posed in this paper. On the other hand, the fact that CGAN-local achieves the second best performance on a majority of datasets and metrics suggests that even without the BundleNet framework, there is value to partitioning a distribution into neighborhoods and conditioning on each. We believe our results suggest that existing generative architectures cannot satisfactorily solve the problem of learning fibers.

## 6 DISCUSSION

**Topological priors on fibers:** As mentioned in section 4, BundleNet requires one to choose an underlying fiber distribution $\mathcal{D}_Z$. The natural choice would be to use Gaussian distributions since they are tractable computationally and to some degree universal in statistics. A natural question is: how well can a Gaussian prior capture more topologically diverse fibers—for example, circular fibers that arise in tasks that involve periodic systems or invariance to 1-dimensional rotations?

To test this, we trained an invertible neural network to map a 1-dimensional Gaussian, a 2-dimensional Gaussian, and a uniform distribution on a circle to a distribution on an oval in the plane. In each case the model was the same deep invertible architecture with RNVP blocks trained with (forward and backward) KL-divergence. Visualizations of the result can be found in figure 3. As expected the circular distribution quickly converges to the target oval while neither the 1- nor 2-dimensional Gaussian converges to a distribution that is visually similar to the oval. This aligns with the basic fact from topology that there is no continuous bijective map from the interval $[0, 1]$ to the circle $\mathbb{S}^1$ (Lee, 2010). Based on this observation and the fact that up to homeomorphism the circle and $\mathbb{R}$ are the only two real, connected, 1-dimensional manifolds, we investigated which prior would give better results on our synthetic datasets.

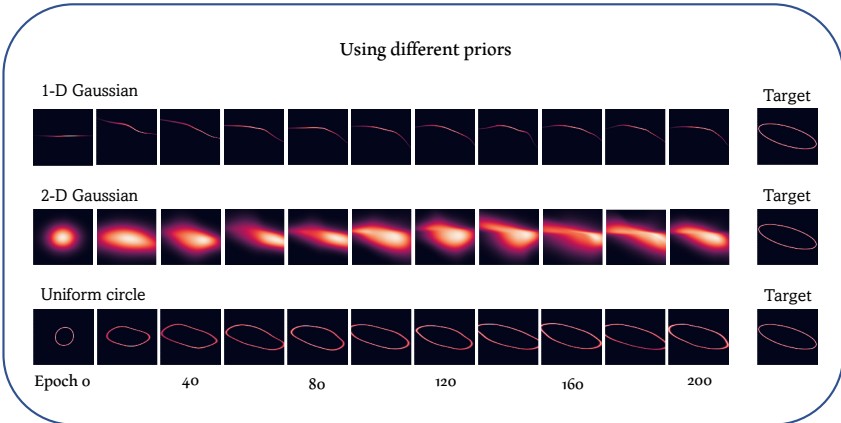

Figure 3: Visualization of the convergence of three different priors to a distribution on an oval: a 1-dimensional Gaussian (**Above**), a 2-dimensional Gaussian (**Middle**), and a uniform distribution on a circle (**Below**)).

Table 3: The Wasserstein-1 metric for fiberwise data generation for models using either a single 1-dimensional Gaussian or a uniform distribution on the circle as a prior.

|  | Gaussian prior | Circular prior |
|---|---|---|
| Torus (global) | $1.640 \pm 0.020$ | $\mathbf{1.338 \pm 0.005}$ |
| Möbius band (global) | $0.327 \pm 0.006$ | $\mathbf{0.240 \pm 0.007}$ |
| Torus (fiberwise) | $1.781 \pm 0.094$ | $\mathbf{1.312 \pm 0.008}$ |
| Möbius band (fiberwise) | $0.411 \pm 0.029$ | $\mathbf{0.231 \pm 0.016}$ |

One might expect that the model with Gaussian prior would perform better on the Möbius band, whose fibers are intervals while the circle prior would lead to better performance on the torus whose fibers are circles. Surprisingly, the circle prior resulted in better performance on both the torus and Möbius band. A possible explanation for this is that the circle prior is a uniform distribution supported on a compact set. The same is true for the fibers in the Möbius band and it is possible that the model has a harder time concentrating the density of a Gaussian distribution than collapsing the shape of a circle to (approximately) a line segment. On the other hand, data collected in Appendix section A.3.2 shows that in some cases the effect of prior choice is negligible. In the end, all of our experiments reinforce the conclusion that while specific prior choices may be important to achieving optimal behavior, our model is relatively robust to this hyperparameter.

**Fiber topology need not be uniform:** While the theory of fiber bundles is rich and well-studied in math, it is perhaps restrictive to have to assume that the data that arises in the real world has this very particular and highly-structured form. Our results on the non-synthetic datasets shown in tables 1 and 2, however suggest that the neighborhood-based approach allows BundleNet to be more flexible. To test whether BundleNets can perform well on datasets whose fibers are not all the same (specifically homeomorphic) we train and test it on a new dataset called the *sliced torus* that is identical to our torus dataset except that the fiber varies continuously from a point to a full circle and back as we traverse the ring.

The quantitative results of our experiments can be found in table 4 and visualizations can be found in figure 4. We note that WGAN and CGAN both struggle to even produce the general shape of the torus, while CGAN-local gets the torus shape mostly right but clumps points from each neighborhood together. Both BundleNet and CGAN-local have trouble with the singularity where the fiber becomes a point (though BundleNet appears to handle this better). For reference, each fiber consists of 100 points. In most cases we see that generative metrics are comparable to those we saw on the torus dataset and almost all cases (except global $\mathcal{W}_2$ and fiberwise MSMD) the results on the sliced torus are better than the next best model we tested on the torus dataset. This, coupled with our superior performance on real-world data that is unlikely to take the form of a true

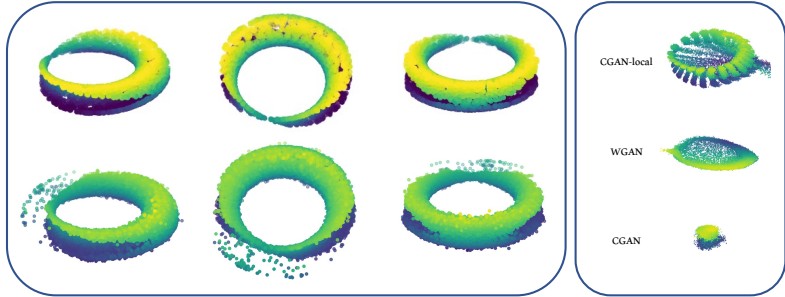

Figure 4: Model performance on a "bundle" where the fibers have different geometry and topology. (**Left, Top row**) Three different views of the training distribution for the sliced torus. (**Left, Bottom row**) Three different views of a 70,000 global points generated by a BundleNet after training. (**Right**) Reconstructions of the the sliced torus using other models.

Table 4: Torus vs sliced torus. Torus results are juxtaposed with those of the sliced torus to give an approximate benchmark for comparison since both distributions have similar scale.

|  | Torus (global) | Sliced (global) | Torus (fiberwise) | Sliced (fiberwise) |
|---|---|---|---|---|
| MSMD | $0.073 \pm 0.001$ | $\mathbf{0.051 \pm 0.001}$ | $0.017 \pm 0.002$ | $\mathbf{0.006 \pm 0.001}$ |
| MMD ($\times 10^{-3}$) | $\mathbf{0.406 \pm 0.018}$ | $2.127 \pm 0.071$ | $\mathbf{10.02 \pm 0.557}$ | $\mathbf{10.361 \pm 0.764}$ |
| $\mathcal{W}_1$ | $\mathbf{0.461 \pm 0.020}$ | $1.243 \pm 0.026$ | $0.251 \pm 0.013$ | $\mathbf{0.195 \pm 0.008}$ |
| $\mathcal{W}_2$ | $\mathbf{0.793 \pm 0.108}$ | $4.813 \pm 0.110$ | $\mathbf{0.071 \pm 0.008}$ | $\mathbf{0.069 \pm 0.006}$ |

fiber bundle, suggest that although our architecture is formulated to deal with situations where the fibers are topologically homogeneous, BundleNets retains sufficient expressivity to allow for more complicated relationships in the data.

Both this section and the previous one investigate topological effects on model performance. It is hard to extrapolate too broadly from such small synthetic experiments, but our results do suggest that exploring latent space topology might be an interesting direction for further research.

### 6.1 LIMITATIONS

While the generative results shown above suggest that BundleNet is a powerful way to learn fibers, it has some drawbacks compared to other models we tested. The most glaring of these is (1) slow training speed, which arises from our implementation of KL-divergence, which has computational complexity $\mathcal{O}(n^2)$ where $n$ is the size of a typical neighborhood and (2) memory considerations when the data is high dimensional, since the default implementation attempts to compute loss terms using the entire neighborhood. A potential solution to both of these problems, would be to subsample from a neighborhood. We also found that, in its current form, BundleNet was not particularly well-suited to image data (partially because of dimensionality considerations). We believe adapting the BundleNet framework to high dimensional data should be the top priority for future work.

## 7 CONCLUSION

Fibers are a useful lens through which to understand a machine learning task. However, they have been little explored in the deep learning literature so far. In this paper we showed that several promising, existing, candidate models struggled to consistently model fibers for both real and synthetic datasets. Drawing on deep ideas from topology, we introduced a new model architecture, the Bundle Network, which by design can model the fibers of a task by breaking the input space into neighborhoods and learning a function on each that decomposes input into a product of its label and all additional variation in the data. We hope this opens a door, both for applying methods from differential topology to machine learning and for further explorations into methods that can help data scientist better model and understand the fibers of their datasets.

## 8 REPRODUCIBILITY STATEMENT

In the interest of making our results reproducible and able to be easily expanded upon, we make our codebase available to the public, including our implementations of BundleNet as well as the WGAN, CGAN, and CGAN-local models we used. We also include training scripts for different models with sensible defaults as well as an evaluation script and examples. Finally, we provide the data used throughout this paper including our specific splits. All of this code and data can be found in our repository at `https://github.com/NicoCourts/bundle-networks/`.

### ACKNOWLEDGMENTS

This research was supported by the Mathematics for Artificial Reasoning in Science (MARS) initiative via the Laboratory Directed Research and Development (LDRD) investments at Pacific Northwest National Laboratory (PNNL). PNNL is a multi-program national laboratory operated for the U.S. Department of Energy (DOE) by Battelle Memorial Institute under Contract No. DE-AC05-76RL0-1830.

The authors would also like to thank Keerti Kappagantula and WoongJo Choi for introducing them to the material science problems that inspired this work.

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

## A  APPENDIX

### A.1  DATASETS

In this section we document our method for either generating or modifying datasets used in this work.

**Torus:** This is a purely synthetic dataset that uses an embedding of a torus (a 2-dimensional surface/manifold) in $\mathbb{R}^3$. Given two radii $r$ and $R$, the torus parameterization is given by:

$$T(r, R) = \{((R + r\cos\theta)\cos\phi, (R + r\cos\theta)\sin\phi, r\sin\theta) : 0 \leq \phi, \theta < 2\pi\}.$$

Our data uses $R = 8$ and $r = 2$ and consists of 1000 points generated by uniformly sampling parameters $\phi$ and $\theta$ from $[0, 2\pi)$. Notice that this does not represent a uniform sampling from the surface itself since it more heavily represents the inner ring of the torus. When we evaluate, we use rejection sampling to sample uniformly from the surface. The base space $Y$ is given by a circle of radius $4.5$.

**Möbius band:** This is our second synthetic data and another surface in $\mathbb{R}^3$. This one is given by the parameterization:

$$M(r, R) = \{(R\cos\theta, R\sin\theta, 0) + s \cdot A_\theta B_{\frac{\theta}{2}} A_\theta^{-1}(-\cos\theta, -\sin\theta, 0) : 0 \leq \theta < 2\pi, -r \leq s \leq r\}$$

where $A_\theta$ is the rotation matrix around the $z$-axis through the angle $\theta$ and $B_\theta$ is the rotation matrix around the $x$ axis through the angle $\theta$. Our data (both for training and evaluation) consists of 1000 points with $r = 2$ and $R = 8$ which are sampled uniformly over the parameters $\theta$ and $s$. The base space $Y$ is a radius $4.5$ circle.

The two real-world datasets, **Wine Quality** (Cortez et al., 2009) and **Airfoil Noise** (Brooks et al., 1989), are described more completely on their pages in the UCI ML repository (Dua & Graff, 2017). The Wine Quality dataset is provided as two files (for red and white wine) with the intention of predicting quality from 11 measured quantities. We combine these two sets into one so that we have 11 inputs and two (both categorical) outputs: color and quality. The Airfoil Noise dataset has 5 input parameters and one output parameter (scaled sound pressure level), which we left unaltered. As a pre-processing step we normalized all parameters in each dataset to span the range 0 to 10 by applying $x \mapsto \frac{10}{M}(x-m)$ where $m$ and $M$ are the minimum and maximum values of each parameter. We made an 80/20% train/test split for each.

## A.2 Loss functions and evaluation metrics

There has been much work in the world of GANs to establish metrics that can quantify the quality of a generative model. This includes the widely-used Inception Score (Salimans et al., 2016) and Fréchet Inception Distance (Heusel et al., 2017). Many of these metrics, however, rely on the assumption that the data has the structure of an image. For instance, in both the methods mentioned above one computes these metrics by using a pretrained Inception v.3 model (Szegedy et al., 2015) to extract latent features from generated images. One then uses methods from statistics/optimal transport like the Wasserstein distance or Kullback-Leibler divergence to evaluate similarity under the assumption that data in these latent spaces follow Gaussian distributions.

Since the data we are using are not images, and since the assumption that our data follows some Gaussian distribution is catastrophically restrictive, we choose some standard two-sample tests to quantitatively compare the performance of different models. One could make the argument that with the blessing of low-dimensionality we are able to more directly access the metrics that are only being approximated in methods such as those for images that rely on a pretrained classifier model.

**Mean squared minimum distance:** The simplest of our metrics, the mean squared minimum distance (MSMD) takes two samples $S_1$ and $S_2$ and computes

$$\text{MSMD}(S_1, S_2) = \frac{1}{|S_1|} \sum_{x_1 \in S_1} \min_{x_2 \in S_2} \|x_1 - x_2\|^2.$$

This naïve approach is not as robust to different distributions as the methods that follow, but in the case that $S_2$ is a collection of points uniformly distributed on some known space (e.g. a circle in space corresponding to the true fiber over a point), this number is more readily interpretable as an approximate measure of how "realistic" the points in $S_1$ are, given the region containing $S_2$.

**Kullback-Leibler divergence:** The KL-divergence, or *relative entropy* was a test introduced by Kullback & Leibler (1951) to apply an information-theoretic approach to the problem of detecting *divergence* of two samples by attempting to quantify the amount of signal (or information) that one could use to distinguish between the two. For instance, given two distributions $P$ and $Q$ on $\mathbb{R}$ with densities $p$ and $q$, this quantity is given by

$$D_{\text{KL}}(P||Q) = \int_{-\infty}^{\infty} p(x) \log\left(\frac{p(x)}{q(x)}\right) \, \mathrm{d}x.$$

A problem that needs to be solved to apply the KL-divergence to our samples is to fix a method of density estimation. A suggestion by Wang et al. (2009) is to (for some integer hyperparameter $k$) approximate the density using the $k^{\text{th}}$ nearest neighbor of a point in each population. This approximation is fast and easily made to be differentiable and for this reason we use it both in our loss function and as an evaluation metric.

Note that the KL-divergence is not an honest metric and in particular is *not symmetric*. The forward divergence (KL-fwd) is $D_{\text{KL}}(P||Q)$ where $P$ is the true/target distribution and the backward divergence (KL-bwd) is $D_{\text{KL}}(Q||P)$. One way of thinking of this difference is that $D_{\text{KL}}(P||Q)$

measures how well $Q$ covers $P$ and vice versa. We include both separately in our results to give a holistic view of the performance.

**Wasserstein metric:** The Wasserstein family of metrics discovered in many works including (Kantorovich, 1940) and (Vaserstein, 1969) are based on the idea of measuring the amount of work that would need to be done to transport the (probability) mass of one distribution to another. Let $(\mathcal{X}, d_{\mathcal{X}})$ be a metric space and $\mu, \nu$ probability measures on $\mathcal{X}$. For any $p > 0$, one defines the $p$-Wasserstein distance between these two distributions to be

$$
W_p(\mu, \nu) = \left( \inf_{\gamma \in \Gamma(\mu, \nu)} \int_{(x,y) \in \mathcal{X} \times \mathcal{X}} d_{\mathcal{X}}(x, y)^p \, \mathrm{d}\gamma(x, y) \right)^{1/p}
$$

where $\Gamma(\mu, \nu)$ denotes the space of probability measures on $\mathcal{X} \times \mathcal{X}$ with margins $\mu$ and $\nu$. To implement this, we piggyback off the work of (Feydy et al., 2019), who provide a wonderful library `geomloss` of differentiable PyTorch implementations of many two-sample distances.

**Maximum mean discrepancy:** First introduced in (Gretton et al., 2012), the maximum mean discrepancy (MMD) between two samples is computed by appealing to a dual problem: consider a collection of functions $\mathcal{F}$ from our probability space $\mathcal{X}$ to $\mathbb{R}$. Then the MMD can be thought of as an expression of how "different" two distributions can look under different choices of $f \in \mathcal{F}$. More concretely,

$$
\mathrm{MMD}(\mathcal{F}; P, Q) = \sup_{f \in \mathcal{F}} \left( \mathbb{E}_{x \sim P}[f(x)] - \mathbb{E}_{y \sim Q}[f(y)] \right).
$$

An implementation of MMD can be found in `geomloss` (Feydy et al., 2019) using a single Gaussian kernel.

## A.3 HYPERPARAMETERS

Below we document our choices of hyperparameters for our models. 'Neural net depth/width' describes the structure of the MLP that is used in each model (although in subtly different ways). For our model, these parameters describe the neural nets that predict the weights and biases for our RNVP blocks. For the GAN models, these parameters control the size/shape of the generator and discriminator. 'Num. neighborhoods' is only applicable in models that compute neighborhoods to condition upon (i.e. BundleNet and CGAN-nbhd). 'Num. invertible blocks' gives the number of RNVP blocks and only applies to our architecture. 'Num. circular priors' again only applies to our model and implicitly defines the number of Gaussian priors we use as well. The formula is $B + 2C + G = D$ where $C$ is the number of circles, $B$ the dimension of the output data, $G$ the number of Gaussian priors, and $D$ the total/latent dimension.

Mathematically, the combination of one-dimensional priors represents a (tensor) product of distributions, but in practice this has a simple form. For each neighborhood, data from the bundle is pushed forward through the encoder to get sample statistics including the coordinate-wise means and standard deviations as well as the mean and standard deviation of the magnitudes for each circular prior. These sample statistics are used to pick a radius for each circular prior and to reparameterize a standard normal distribution for the Gaussian priors. Every circle is encoded in two parameters as a circle with the radius determined above and every Gaussian (again, formed using sample statistics) occupies a single parameter. The values are sampled independently from their respective distributions and then concatenated into a vector in latent space representing a sample from $\mathcal{D}_Z$.

### A.3.1 EFFECT OF NUMBER OF NEIGHBORHOODS ON GENERATION

In this section we explore how the number of neighborhoods used to partition $Y$ affects the (global and fiberwise) generative performance of BundleNet. We note that for both synthetic datasets the topological character of the inverse images of neighborhoods in input space $X$ changes dramatically as one moves from a single neighborhood (corresponding to a global trivialization) to two or more open neighborhoods. When we use a single neighborhood $U = Y$, then $\pi^{-1}(U)$ is of course a torus or Möbius band respectively. When we use two or more neighborhoods, $\pi^{-1}(U)$ admits a deformation retraction onto the circle $\mathbb{S}^1$ in the case of the torus, and is contractible in the case of the Möbius band. Under the assumption that a space becomes easier to model as its topological complexity decreases, we hypothesized that we would see a significant improvement in performance when moving from 1 to 2 neighborhoods, but would not see substantial improvements afterwards.

Table 5: Hyperparameter choices used in all models trained and evaluated during the course of this work (unless otherwise noted in the main text). If multiple values are given, the format is (torus, Möbius band, wine, airfoil).

MODEL HYPERPARAMETERS

|  | BundleNet (ours) | WGAN | CGAN | CGAN-nbhd |
|---|---|---|---|---|
| Initial learning rate | $10^{-4}$ | $10^{-4}$ | $10^{-4}$ | $10^{-4}$ |
| Latent dimension | $(8, 8, 8, 12)$ | 10 | 10 | 10 |
| Neural net depth. | 5 | 5 | 5 | 5 |
| Neural net width. | 512 | 128-1024 | 128-1024 | 128-1024 |
| Num. neighborhoods | $(25, 25, 13, 25)$ | N/A | N/A | $(25, 25, 13, 25)$ |
| Num. invertible blocks | 5 | N/A | N/A | N/A |
| Num. circular priors | 3 | N/A | N/A | N/A |

Table 6: Generative performance (Wasserstein-1 distance) of BundleNet on the Torus dataset in terms of the number of neighborhoods of $Y$ used. Each model was trained once and evaluated multiple times to generate $95\%$ CIs for a particular set of model weights.

TORUS GENERATION PERFORMANCE BY NUMBER OF NEIGHBORHOODS

| Number of Neighborhoods | Global | Fiberwise |
|---|---|---|
| 1 | $0.579 \pm 0.030$ | $6.059 \pm 0.289$ |
| 2 | $0.575 \pm 0.052$ | $0.282 \pm 0.012$ |
| 5 | $0.573 \pm 0.037$ | $0.303 \pm 0.007$ |
| 10 | $0.660 \pm 0.034$ | $0.425 \pm 0.016$ |
| 25 | $0.573 \pm 0.036$ | $0.230 \pm 0.017$ |
| 100 | $0.632 \pm 0.028$ | $0.439 \pm 0.024$ |

The results of our tests (found in table 6) fit well with our observations about topological complexity. We find that global generative performance seems not to be strongly tied to the number of neighborhoods but that fiberwise performance increases dramatically as one moves from 1 to 2 neighborhoods. For any number of neighborhoods greater than 2 we do not see a particular improvement to either global or fiberwise performance. These results suggest that the use of local trivializations is mainly helpful for the task of modeling fibers themselves, but may not be as critical to modeling the distribution as a whole. Note that this is somewhat in contrast to the improved performance of CGAN-local over CGAN, which indicated that at least within the GAN framework, use of local neighborhoods was helpful for both global and fiberwise metrics.

A.3.2 EFFECT OF TOPOLOGICAL PRIOR CHOICE ON MODEL PERFORMANCE

Table 7: Generative performance (Wasserstein-1 distance) of BundleNet on the Airfoil dataset in terms of the number of circular priors used in latent space. The total dimension of the latent space was 18. 1-dimensional Gaussians account for additional dimensions where circle priors were not used.

AIRFOIL GENERATION PERFORMANCE BY NUMBER OF CIRCULAR PRIORS

| Number of Circular Priors | Global | Fiberwise |
|---|---|---|
| 0 | $1.283 \pm 0.013$ | $2.943 \pm 0.056$ |
| 2 | $1.037 \pm 0.008$ | $2.899 \pm 0.095$ |
| 4 | $1.172 \pm 0.014$ | $2.925 \pm 0.093$ |
| 6 | $1.124 \pm 0.007$ | $3.072 \pm 0.072$ |

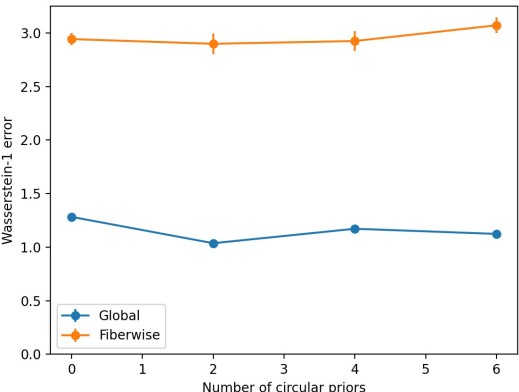

Figure 5: Effect on generation error (Wasserstein-1 distance) on the Airfoil Noise dataset by number of circular priors. Error bars represent 95% CIs.

In this section we examine how the choice of topology for the priors (circles vs. Gaussians) affects generative performance in the wild. Recall that we found a slight advantage to using circles when exploring our synthetic datasets in section 6. We trained BundleNet using a 18-dimensional latent space with different numbers of circular priors (with the rest of the parameters one-dimensional Gaussians). The results can be found in table 7 and figure 5. For the most part, these results show that Airfoil Noise generative capacity is not strongly affected by the number of circular priors chosen. We hypothesize that this is due to two factors: first, this dataset is a difficult one (for instance, we see that the margins between different models are very slim) and second, there are no parameters in this dataset that are naturally modeled as a circle. The fact that we saw better performance on our synthetic datasets suggests that there is still value to tuning this hyperparameter depending on the data under consideration. We believe that the relative invariance BundleNet shows to choice of priors speaks to the robustness of our model to a wide array of hyperparameter choices.

### A.4  A COMPARISON BUNDLENET AND OTHER MODELS

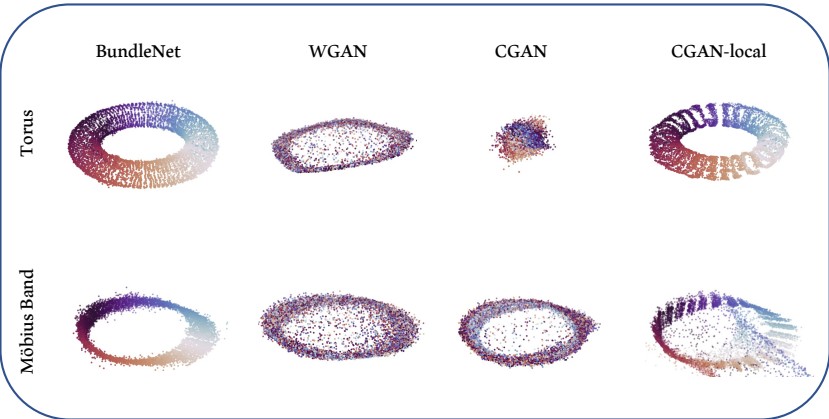

Figure 6: Visualizations of synthetic dataset reconstruction for each model type. To produce each image, 100 evenly distributed base points were chosen and we generated 200 points in the fiber over each (where such capability is available). The colors correspond to the angle parameterizing the circle in the base space. Note that we do not expect the colors in WGAN's reconstruction to be well-organized since this model has no way of consistently modeling specific fibers like the other models do.

We begin this section with a visual and qualitative analysis of the performance of BundleNet, WGAN, CGAN, and CGAN-local models on our synthetic datasets. The images in this section are restricted by being 2D projections of 3D point clouds, but they should serve to help illustrate why the tables in section 5.2 and section A.5 appear as they do.

Perhaps the most immediate observation that can be made is that CGAN suffers a complete collapse on the torus dataset. This corresponds to high loss in both the global and fiberwise regimes (it is the worst-performing model on this dataset). CGAN does considerably better on the Möbius band dataset (it is the second-best performing according to the Wasserstein-1 metric, c.f. table 1). Even in the case of good global generation metrics, however, CGAN fails to capture fiber structure, as can be seen in the way that different colors which should be localized to specific sections of the band, are distributed throughout the reconstruction. We discuss this failure more carefully below in section A.4.1. We also discuss how we attempted to train CGAN in different ways so as to make it more amenable to fiber discovery.

Of all the GAN architectures, the reconstructions generated by CGAN-local appear to be the most faithful to the true distribution. Nevertheless, one can detect ways that CGAN-local's reconstructions deviate from the torus and Möbius band respectively. In both reconstructions, CGAN-local tends to cluster neighborhoods together giving the resulting visualization a "banded" quality. Note that the bands in the Bundlenet torus are not due to a similar failing, but are actually visible individual fibers (100 evenly-distributed fibers around the base circle). This behavior could account for CGAN-local's (relatively) poor global generative performance in the main results. We also see an intriguing failure in CGAN-local's reconstruction of the Möbius band where some of the fibers are smeared outside of the band's bounds. We speculate that this smearing occurs because CGAN-local struggles to capture the twist in the band. This can also be seen in the sliced torus in figure 4, which we analogously hypothesized was related to the singularity present where the fiber becomes a point. An interesting avenue for further research is how topology affects the ability of a generative model to approximate a distribution.

WGAN is somewhat unique among tested architectures in that it does not incorporate a conditional aspect to learn fibers over a point. As such, the point clouds generated by WGAN map basepoints uniformly throughout the generated data. WGAN has a difficult time recreating the circular fibers for the torus dataset, but gives a reasonable reconstruction of the Möbius band. Again, these visualizations align with our quantitative results.

### A.4.1 CGAN, FIBERS, AND TRAINING

Table 8: Generative performance metrics for the torus dataset for CGAN variations

GLOBAL

|  | BundleNet (ours) | CGAN | CGAN-local-mixed | CGAN-local |
|---|---|---|---|---|
| MSMD | $\mathbf{0.073 \pm 0.001}$ | $36.41 \pm 0.365$ | $58.63 \pm 0.355$ | $0.930 \pm 0.013$ |
| MMD ($\times 10^{-3}$) | $\mathbf{0.406 \pm 0.018}$ | $35.04 \pm 0.214$ | $173.4 \pm 1.173$ | $2.336 \pm 0.054$ |
| KL-Fwd | $\mathbf{0.812 \pm 0.035}$ | $10.62 \pm 0.021$ | $11.41 \pm 0.023$ | $3.615 \pm 0.028$ |
| KL-Bwd | $\mathbf{0.221 \pm 0.021}$ | $11.97 \pm 0.013$ | $15.78 \pm 0.010$ | $4.070 \pm 0.044$ |
| $\mathcal{W}_1$ | $\mathbf{0.461 \pm 0.020}$ | $7.114 \pm 0.015$ | $7.821 \pm 0.017$ | $1.610 \pm 0.031$ |
| $\mathcal{W}_2$ | $\mathbf{0.793 \pm 0.108}$ | $25.76 \pm 0.097$ | $31.60 \pm 0.123$ | $2.715 \pm 0.118$ |

FIBERWISE

|  | BundleNet (ours) | CGAN | CGAN-local-mixed | CGAN-local |
|---|---|---|---|---|
| MSMD | $\mathbf{0.017 \pm 0.002}$ | $29.81 \pm 1.911$ | $55.54 \pm 0.370$ | $0.148 \pm 0.011$ |
| MMD ($\times 10^{-3}$) | $\mathbf{10.02 \pm 0.557}$ | $83.46 \pm 1.230$ | $153.8 \pm 1.394$ | $28.74 \pm 1.941$ |
| KL-Fwd | $\mathbf{5.047 \pm 0.130}$ | $16.91 \pm 0.110$ | $18.01 \pm 0.064$ | $7.440 \pm 0.180$ |
| KL-Bwd | $\mathbf{1.907 \pm 0.090}$ | $9.422 \pm 0.059$ | $11.45 \pm 0.040$ | $3.122 \pm 0.194$ |
| $\mathcal{W}_1$ | $\mathbf{0.251 \pm 0.013}$ | $7.996 \pm 0.015$ | $7.980 \pm 0.021$ | $0.450 \pm 0.018$ |
| $\mathcal{W}_2$ | $\mathbf{0.101 \pm 0.010}$ | $32.60 \pm 0.114$ | $32.73 \pm 0.167$ | $0.183 \pm 0.010$ |

In this section we focus on CGAN and its failure modes on our synthetic data. We show how we can improve CGAN by leveraging some of the same methods developed for BundleNets. To this end, we document our attempt to explore two different training strategies for CGAN, which we designate CGAN-local-mixed and CGAN-local. In both incarnations, we compute neighborhoods of $Y$ and their representatives and then use these representatives to condition upon. During training, one of two natural choices can be made: constrain each batch to points from a single neighborhood or sample each batch from across all neighborhoods. We call the former CGAN-local and the latter CGAN-local-mixed. We view the progression from CGAN to CGAN-local-mixed to CGAN-local as capturing the development of a conditional generative architecture that is better suited to the fiber task. Furthermore, we see the poor performance of CGAN-local-mixed as evidence that BundleNet's performance cannot be attributed *simply* to reducing the conditioning set from a continuum to a finite number of neighborhood representatives.

The results in table 8 document the relative performance of these models on the torus dataset. We see that the naïve CGAN is poorly suited to the task and that switching to neighborhood conditioning with CGAN-local-mixed degrades performance across several metrics. On the other hand, restricting each batch to a single neighborhood (CGAN-local) results in a model that performs much more reasonably (but still inferior to) BundleNet. We view CGAN-local as our best effort to equip a GAN with the power to perform the fiber generation task without fundamentally altering the architecture. Thus this serves as our primary point of comparison in the main paper.

## A.5 FULL RESULTS

Below we recreate the full table of results with all metrics tested.

Table 9: Generative performance metrics (global). Each metric is applied to a trained model as detailed in section 5.1 with 95% confidence intervals determined by bootstrapping. Conditioning, when built into the model, is used to generate points over (uniformly sampled) base points to construct the point cloud.

| TORUS | | | | |
|---|---|---|---|---|
| | BundleNet (ours) | WGAN | CGAN | CGAN-local |
| MSMD | $\mathbf{0.073 \pm 0.001}$ | $4.858 \pm 0.042$ | $36.41 \pm 0.365$ | $0.930 \pm 0.013$ |
| MMD ($\times 10^{-3}$) | $\mathbf{0.406 \pm 0.018}$ | $3.832 \pm 0.039$ | $35.04 \pm 0.214$ | $2.336 \pm 0.054$ |
| KL-Fwd | $\mathbf{0.812 \pm 0.035}$ | $6.300 \pm 0.035$ | $10.62 \pm 0.021$ | $3.615 \pm 0.028$ |
| KL-Bwd | $\mathbf{0.221 \pm 0.021}$ | $5.388 \pm 0.042$ | $11.97 \pm 0.013$ | $4.070 \pm 0.044$ |
| $\mathcal{W}_1$ | $\mathbf{0.461 \pm 0.020}$ | $2.191 \pm 0.011$ | $7.114 \pm 0.015$ | $1.610 \pm 0.031$ |
| $\mathcal{W}_2$ | $\mathbf{0.793 \pm 0.108}$ | $3.442 \pm 0.033$ | $25.76 \pm 0.097$ | $2.715 \pm 0.118$ |

| MÖBIUS BAND | | | | |
|---|---|---|---|---|
| | BundleNet (ours) | WGAN | CGAN | CGAN-local |
| MSMD | $\mathbf{0.029 \pm 0.001}$ | $3.187 \pm 0.026$ | $1.167 \pm 0.016$ | $0.404 \pm 0.007$ |
| MMD ($\times 10^{-3}$) | $\mathbf{0.509 \pm 0.026}$ | $5.875 \pm 0.038$ | $5.909 \pm 0.031$ | $4.236 \pm 0.045$ |
| KL-Fwd | $\mathbf{1.031 \pm 0.017}$ | $7.601 \pm 0.040$ | $5.936 \pm 0.023$ | $4.686 \pm 0.041$ |
| KL-Bwd | $\mathbf{0.002 \pm 0.018}$ | $7.174 \pm 0.044$ | $5.832 \pm 0.040$ | $6.723 \pm 0.054$ |
| $\mathcal{W}_1$ | $\mathbf{0.264 \pm 0.004}$ | $2.048 \pm 0.013$ | $1.844 \pm 0.015$ | $6.497 \pm 0.101$ |
| $\mathcal{W}_2$ | $\mathbf{0.060 \pm 0.009}$ | $2.829 \pm 0.053$ | $2.384 \pm 0.051$ | $40.63 \pm 0.814$ |

WINE QUALITY

|  | BundleNet (ours) | WGAN | CGAN | CGAN-local |
|---|---|---|---|---|
| MSMD | $\mathbf{1.581 \pm 0.023}$ | $68.26 \pm 1.436$ | $6.630 \pm 0.084$ | $3.275 \pm 0.042$ |
| MMD $(\times 10^{-3})$ | $\mathbf{0.820 \pm 0.014}$ | $95.59 \pm 1.488$ | $10.67 \pm 0.123$ | $1.492 \pm 0.016$ |
| KL-Fwd | $\mathbf{3.158 \pm 0.062}$ | $24.74 \pm 0.142$ | $9.966 \pm 0.114$ | $6.393 \pm 0.054$ |
| KL-Bwd | $\mathbf{2.902 \pm 0.120}$ | $115.7 \pm 0.336$ | $19.28 \pm 0.132$ | $10.067 \pm 0.097$ |
| $\mathcal{W}_1$ | $\mathbf{1.733 \pm 0.011}$ | $99.22 \pm 0.392$ | $4.054 \pm 0.014$ | $2.114 \pm 0.011$ |
| $\mathcal{W}_2$ | $\mathbf{2.155 \pm 0.036}$ | $6271. \pm 40.40$ | $8.950 \pm 0.066$ | $2.567 \pm 0.028$ |

AIRFOIL NOISE

|  | BundleNet (ours) | WGAN | CGAN | CGAN-local |
|---|---|---|---|---|
| MSMD | $\mathbf{0.512 \pm 0.022}$ | $24.54 \pm 0.356$ | $3.474 \pm 0.057$ | $1.300 \pm 0.030$ |
| MMD $(\times 10^{-3})$ | $\mathbf{1.840 \pm 0.013}$ | $38.86 \pm 0.480$ | $4.222 \pm 0.044$ | $2.742 \pm 0.066$ |
| KL-Fwd | $\mathbf{5.810 \pm 0.089}$ | $14.87 \pm 0.059$ | $9.131 \pm 0.064$ | $7.247 \pm 0.095$ |
| KL-Bwd | $\mathbf{2.430 \pm 0.018}$ | $15.96 \pm 0.058$ | $6.716 \pm 0.036$ | $4.580 \pm 0.051$ |
| $\mathcal{W}_1$ | $\mathbf{1.448 \pm 0.022}$ | $4.741 \pm 0.034$ | $2.508 \pm 0.025$ | $1.735 \pm 0.013$ |
| $\mathcal{W}_2$ | $\mathbf{1.888 \pm 0.123}$ | $13.95 \pm 0.198$ | $4.964 \pm 0.120$ | $2.121 \pm 0.050$ |

Table 10: Generative performance metrics (fiberwise). Results were obtained by randomly sampling 5 base points and generating points in the fiber over each. Each metric is applied to the generated points and points sampled from the true distribution and averaged. Intervals represent 95% CIs over 10 repeated experiments. WGAN is excluded since it has no built-in way to condition on the base point.

TORUS

|  | BundleNet (ours) | CGAN | CGAN-local |
|---|---|---|---|
| MSMD | $\mathbf{0.017 \pm 0.002}$ | $29.81 \pm 1.911$ | $0.148 \pm 0.011$ |
| MMD $(\times 10^{-3})$ | $\mathbf{10.02 \pm 0.557}$ | $83.46 \pm 1.230$ | $28.74 \pm 1.941$ |
| KL-Fwd | $\mathbf{5.047 \pm 0.130}$ | $16.91 \pm 0.110$ | $7.440 \pm 0.180$ |
| KL-Bwd | $\mathbf{1.907 \pm 0.090}$ | $9.422 \pm 0.059$ | $3.122 \pm 0.194$ |
| $\mathcal{W}_1$ | $\mathbf{0.251 \pm 0.013}$ | $7.996 \pm 0.015$ | $0.450 \pm 0.018$ |
| $\mathcal{W}_2$ | $\mathbf{0.101 \pm 0.010}$ | $32.60 \pm 0.114$ | $0.183 \pm 0.010$ |

MÖBIUS BAND

|  | BundleNet (ours) | CGAN | CGAN-local |
|---|---|---|---|
| MSMD | $\mathbf{0.009 \pm 0.001}$ | $0.810 \pm 0.125$ | $0.067 \pm 0.011$ |
| MMD $(\times 10^{-3})$ | $\mathbf{20.82 \pm 2.724}$ | $144.0 \pm 0.482$ | $68.85 \pm 5.721$ |
| KL-Fwd | $\mathbf{7.646 \pm 0.125}$ | $14.31 \pm 0.146$ | $10.53 \pm 0.278$ |
| KL-Bwd | $\mathbf{1.856 \pm 0.106}$ | $8.566 \pm 0.068$ | $5.507 \pm 0.158$ |
| $\mathcal{W}_1$ | $\mathbf{0.279 \pm 0.011}$ | $9.860 \pm 0.038$ | $8.228 \pm 0.904$ |
| $\mathcal{W}_2$ | $\mathbf{0.071 \pm 0.008}$ | $58.47 \pm 0.364$ | $193.8 \pm 38.25$ |

WINE QUALITY

|  | BundleNet (ours) | CGAN | CGAN-local |
|---|---|---|---|
| MSMD | $\mathbf{3.234 \pm 0.883}$ | $10.70 \pm 1.703$ | $5.581 \pm 0.974$ |
| MMD $(\times 10^{-3})$ | $\mathbf{21.84 \pm 13.08}$ | $\mathbf{48.37 \pm 40.40}$ | $71.77 \pm 30.48$ |
| KL-Fwd | $\mathbf{-0.091 \pm 1.186}$ | $5.942 \pm 1.303$ | $2.215 \pm 0.848$ |
| KL-Bwd | $\mathbf{0.856 \pm 0.474}$ | $10.56 \pm 1.468$ | $7.164 \pm 1.096$ |
| $\mathcal{W}_1$ | $\mathbf{1.917 \pm 0.172}$ | $3.666 \pm 0.233$ | $2.926 \pm 0.169$ |
| $\mathcal{W}_2$ | $\mathbf{2.407 \pm 0.763}$ | $7.941 \pm 0.951$ | $5.263 \pm 0.634$ |

AIRFOIL NOISE

| | BundleNet (ours) | CGAN | CGAN-local |
|---|---|---|---|
| MSMD | $\mathbf{1.920 \pm 0.157}$ | $2.793 \pm 0.367$ | $\mathbf{2.134 \pm 0.190}$ |
| MMD ($\times 10^{-3}$) | $\mathbf{35.872 \pm 1.253}$ | $\mathbf{37.37 \pm 0.935}$ | $37.96 \pm 0.714$ |
| KL-Fwd | $\mathbf{-2.694 \pm 0.430}$ | $-1.259 \pm 0.543$ | $\mathbf{-2.121 \pm 0.229}$ |
| KL-Bwd | $\mathbf{3.250 \pm 0.085}$ | $4.908 \pm 0.175$ | $4.863 \pm 0.054$ |
| $\mathcal{W}_1$ | $\mathbf{3.124 \pm 0.089}$ | $3.563 \pm 0.158$ | $\mathbf{3.076 \pm 0.063}$ |
| $\mathcal{W}_2$ | $\mathbf{6.744 \pm 0.459}$ | $8.054 \pm 0.711$ | $\mathbf{6.394 \pm 0.261}$ |

