# OpenReview forum: "Bundle Networks: Fiber Bundles, Local Trivializations, and a Generative Approach to Exploring Many-to-one Maps"
_ICLR.cc/2022/Conference — ICLR 2022 Poster_

### Official Review · Reviewer_Hz2R · 2021-11-02

**Correctness:** 3
**Technical Novelty And Significance:** 3
**Empirical Novelty And Significance:** 3
**Recommendation:** 6
**Confidence:** 3

**Main Review:**

[Strengths]
1) (A new problem) The formalization of a new machine learning problem, i.e., the problem of learning fibers of a ML task.
2) (A new solution) An effective scheme is proposed to solve the fiber learning problem with encouraging results.


[Weakness]
1) (The motivation). The proposed fiber learning problem is interesting and novel to me. However, the rationality of modeling the many-to-one relationship via the strict "product structure" seems not sufficiently explained. It is suggested to give more vivid examples in CV/ML to show the model's suitability.
2) (Conditioning may weaken the expressive power). To avoid training a different model for each $U_i$ in $\mathcal{U}$, conditioning is used in the proposed BundleNet. However, using conditioning instead of optimizing may limit the model's expressive power.
3) (Clarity of introduction to BundleNet). The description of BundleNet seems not sufficiently clear. I find it a bit difficult to understand the whole process. For example, "elements of $\mathcal{R}$ are in bijection with neighborhoods contained in $\mathcal{U}$ with the objective of having $\Phi_i$ play the role of $\psi_i^{-1}$". It is suggested to rewrite this section for better readability.
4) (Typos). There are typos in this paper. For example, "Finally, each measurable subset $U\in Y$"  --> "$U \subset Y$".





**Summary Of The Paper:**

This paper proposes the Bundle Networks as a framework to learn many-to-one maps. The empirical performance is promising on two synthetic datasets and two real datasets.
The main contribustions are three-fold:
1) The problem of learning the fibers of a ML task is formalized.
2) An approach is proposed to learn the fibers.
3) A family of deep generative models called a Bundle Network is designed.


**Summary Of The Review:**

This paper proposes a new problem, i.e., the fiber learning problem of a ML task, as well as an effective solution to it. The proposal is novel and interesting to me. However, the motivation, the clearity, and the empirical support can be improved.

---

> ### Author Response · Authors · 2021-11-19
> **Response**
>
> We would like to thank the reviewer for their helpful comments. We are also glad that they found our work interesting. Below we have tried to address some of the reviewer’s concerns from the ‘Weaknesses’ section of their review.
>
> - The reviewer brings up an interesting point about the suitability of the product structure. As another reviewer pointed out, it is not reasonable to expect that the underlying data distribution actually has local trivializations  “on the nose”. We added a note on this in Section 3.1. Rather, just as we always try to approximate nonlinear problems by linear ones when possible, it makes sense to try to approximate a space as a product. If such an approximation is successful, it can have many beneficial downstream effects since so many algorithms generalize nicely to product spaces. Further, your question about CV applications is a great one. Adapting our methods to higher dimensional data is the next priority of our research, but for this paper we chose to lay the groundwork both in terms of theory and some basic examples.
> - The reviewer also brings up an interesting point related to expressivity and conditioning. During model development we wanted to be sure our proposed architecture could potentially scale to a large number of weights and a large number of neighborhoods. Using a different model for each neighborhood seemed problematic from this standpoint. On the other hand, training a different model for each neighborhood and comparing that with our existing results would be interesting since it might help answer a few questions: (1) is model expressivity a limitation within our framework and (2) is there any generalization benefit that a model receives from being able to share weights across many different neighborhoods? This would be an interesting experiment to run in the future.
> - We appreciate the reviewer pointing out that the beginning of Section 4 was not clear. We have revised the first paragraph of this section and hope that this will make the passage easier to read for future readers.
> - We thank the reviewer for catching some of our typos. We have corrected the one pointed out (U∈Y --> U⊂Y) and have tried to eliminate others where we were able to locate them.
>
> Again, we would like to thank the reviewer for taking the time to read through our manuscript and offer useful feedback.

---

### Official Review · Reviewer_tPZF · 2021-11-02

**Correctness:** 4
**Technical Novelty And Significance:** 4
**Empirical Novelty And Significance:** Not applicable
**Recommendation:** 6
**Confidence:** 3

**Main Review:**

*Pro*

+ The paper is clearly written and easy to follow. The figures are of high quality.

+ The method outperforms the baselines at the studied generative modeling tasks.

+ The method is based on a standard construction in differential topology. As such the underlying concepts are mathematically well-founded. Bringing these ideas to machine learning is quite interesting on its own.

*Contra*

- The labelling function on real world data is typically not a fiber bundle $Z\to \operatorname{supp}(\mathcal{D_X}) \stackrel{\pi}{\to} Y$. In other words, the local trivialization the networks tries to learn do not exist.
Admittedly, this limitation is discussed towards the end of the paper in section 6. However, from my point of view, this should be stated more prominently already in section 3.1, when introducing the general idea. Also, it's theoretical implication should have been explored to some extent.

  The empirical examination of this issue in section 6 is insufficient. The topological space which is studied therein, is locally homeomorphic to $\mathbb R \times \mathbb S^1$ except on a measure zero set (with respect to the uniform distribution). So it is unsurprising, that the algorithm is able to learn this space. Overall, the setting is too simple to transfer to real world data, whose topology usually is much more difficult.

  I also want to remark, that Figure 4 reveals problems with predicting the geometry (opposed to the topology) close to the singularity. Do the baselines (WGAN, CGAN) suffer from similar artifacts? Could you provide similar plots as Figure 4 for them?

- The method introduces new hyperparameters, in particular the fiber space $Z$, a probability distribution $\mathcal {D_Z}$ on the fiber and the number $q$ of local homeomorphisms (centroids for k-means). Analysis of the robustness with respect to these choices on real world data is missing.

   Regarding $q$, according to the theory, there should not be an issue, if $q$ is sufficiently large. Is this indeed true? Does choosing $q$ too large lead to a noticable increase of computational cost.

  Regarding $Z$ and $\mathcal{D_Z}$. Choosing these seems not obvious at all on non-synthetic data. Are there any heuristics? Is the prediction robust with respect to these choices? Are there any tradeoffs regading the fiber dimensionality?

- (minor) Figure 6 in Appendix A.4 is not sufficiently described. It is unclear, which points come from training data and which ones come from the reconstruction. Is only the bottom right reconstructed?

**Summary Of The Paper:**

The paper introduces a new architecture for generative modeling, called Bundle Networks. This is based on the mathematical concept of fiber bundles and allows to explore the networks fibers above the labels (their preimages). Training the network then corresponds to learning the local trivialization of the fiber bundle, i.e., of the labeling function.

**Summary Of The Review:**

The idea of the submission is quite interesting and I really like its presentation in the paper. However, the theoretical setting seems not to be applicable to real word data. Furthermore, a study on the robustness with respect to the additional hyperparameters is missing. I therefore give a score of 5.

----

**Update**

I thank the authors for addressing my concerns in the update. It seems that the good performance is rather independent of particular hyperparameter choices and prior distributions, but mostly from the guiding idea of allowing the model to be a *local* (as opposed to *global*) trivialization, as indicated by the experiments with q=1. Yet, a convincing argument is missing. In general, I still think, that there should have been a theoretical study under which conditions and  assumptions on the data Bundle Networks can be successful.

I raise my score to 6.

---

> ### Author Response · Authors · 2021-11-19
> **Response**
>
> We would like to thank the reviewer for investing the time to give us so many helpful comments and suggestions. We have tried to address as many of them as we could within the time frame that we had. Many of those that we were not able to comprehensively address will be on our list for future iterations and follow-ons to our work.
>
> Below we respond to issues brought up by the ‘Contra’ section of the review:
>
> - (First bullet) We agree that it should have been explicitly stated in Section 3.1 that real data (in most cases at least) should not be expected to have an exact local trivialization and that this concept is meant to be an approximation and framework that enables better modeling of fibers. We have added a sentence in the second to last paragraph of Section 3.1 to clarify this.
> - (First bullet, cont.) We also agree that care must be taken with conclusions drawn in Section 6. Our aim was to ask some basic questions and perform simple experiments that could be easily visualized, and which might guide future research. We added a few sentences at the end of Section 6 emphasizing this was only a starting point to asking further questions about the relationship between topology and model performance.
> - (First bullet, cont.) We have included visualizations for other models in figure 4 demonstrating the generative capabilities on the sliced torus dataset. We find that all models have difficulty near the singularity. This is challenging to see for models that generally struggle at reconstructing the distribution, but it is at least clear for CGAN-local where there is a smear at the singularity.
> - (Second bullet) We agreed with the reviewer’s assessment that more study could have been done with respect to hyperparameter choices for BundleNet. We conducted some additional experiments and added new sections in the Appendix (A.3.1 and A.3.2) which summarize our results. A.3.1 studies how the number of neighborhoods effects model performance on the torus and Mobius band. We find that in terms of reconstruction of the global distribution, the model is robust to different numbers of neighborhoods (including only 1). On the other hand, we find that fiberwise performance improves dramatically as one moves from a single neighborhood, to anything more than 1. This may be related to some of the topological changes that occur when moving from one neighborhood (which is the torus or Mobius band respectively) to 2 or more (where neighborhoods become homeomorphic to the circle or a disc). These experiments suggest to us that including the local aspect of local trivialization may be of primary benefit to fiber reconstruction, while being somewhat less important to overall distribution reconstruction.
> - (Second bullet, cont.) In A.3.2 we also analyze some of the topological priors. In particular, we examine the effect of different prior choices on generation quality for the Airfoil Noise dataset. Our findings were that BundleNet is relatively robust to changes in priors on this set, which we expect is due to the fact that this is a difficult dataset with no input parameters that are naturally modelled by circles. We adjusted our language slightly and highlighted that the benefits one may see from tuning this parameter may depend on the data under consideration.
> - Finally, we merged the images in the previous figures 5 and 6 to a single picture with more examples of reconstructions as well as a more complete description of the data pictured. Hopefully this alleviates any confusion.
>
> We would like to thank the reviewer again for their careful reading of our paper. We greatly appreciate their feedback.

---

> > ### Comment · Reviewer_tPZF · 2021-11-28
> > **Update**
> >
> > I thank the authors for addressing my concerns in the update. It seems that the good performance is rather independent of particular hyperparameter choices and prior distributions, but mostly from the guiding idea of allowing the model to be a *local* (as opposed to *global*) trivialization, as indicated by the experiments with q=1. Yet, a convincing argument is missing. In general, I still think, that there should have been a theoretical study under which conditions and  assumptions on the data Bundle Networks can be successful.
> >
> > I raise my score to 6.

---

### Official Review · Reviewer_asbk · 2021-11-02

**Correctness:** 4
**Technical Novelty And Significance:** 4
**Empirical Novelty And Significance:** 3
**Recommendation:** 8
**Confidence:** 3

**Main Review:**

The manuscript is presented clearly and succinctly and is generally well written, and the technical contributions are novel as far as I can tell. I believe this work has potential in influencing future papers and could become important in the future.

When the range is continuous it seems to me that by choosing the clusters to be large or small, one has a trade-off between the "true" fibers for very small clusters, and a regularised version for large ones. It would be interesting to see this discussed in a bit more depth. I would also like to see a little more elaboration on how the number of neighbourhoods for k-means affects the performance, as this seems to affect the overall architecture quite a bit.

It would also add a lot if the task of learning the fiber structure was motivated more by e.g. an application where the fiber structure is helpful in analysing the problem.

My main point of contention is that the datasets considered are very low dimensional, and the manuscript mentions that the model trains very slowly in higher dimensions, but this isn't shown and I suspect that the huge improvement in performance over the baselines would drop off pretty fast as the dimensionality increases. It would be interesting if this could be expanded on (maybe in the appendix).

Overall I think this is a well written and stimulating paper with theoretically motivated technical innovation. The experimental results look impressive besides the lack of any higher dimensional results, but as this is a more theoretical paper, this omission does not hurt the manuscript too much.


**Summary Of The Paper:**

The manuscript introduces Bundle Networks which are neural networks designed with an explicit fiber bundle in the architecture. With this in hand one has an explicit description of the fibers of the fibers associated with a machine learning task. The model is demonstrated on both artificial data, showing that it is capable of learning the fiber bundles where these can be explicitly computed, as well as real world data sets where the fiber bundles are of interest.


**Summary Of The Review:**

I think this is an important addition to the literature and has the possibility of being useful for future work about explainability and disentanglement.

---

> ### Author Response · Authors · 2021-11-19
> **Response**
>
> We are glad that the reviewer found our manuscript to be interesting and we greatly appreciate the helpful suggestions on how to improve it. Below we comment on some of the ways that we have tried to address the reviewer’s comments.
>
> - We agree that understanding how the number of neighborhoods affects model performance (particularly how robust a model is to this hyperparameter) was not satisfactorily addressed in the initial submission. We have added a section (A.3.1) to the Appendix which starts to investigate this. For the two synthetic datasets we find a drastic improvement in performance as we move from a single neighborhood (a global trivialization) to two neighborhoods. We hypothesize that this might be related to the qualitative change in neighborhood topology as we move from one to two or more neighborhoods for the torus and Möbius band. For these simple synthetic datasets we did not find a strong relationship between performance and number of neighborhoods for k > 1. In future work, we would study this more thoroughly on real-world datasets.
> - This paper was actually motivated by a problem in materials science that involves identifying all the ways to process a material so as to achieve a specific material property (material strength for example). While the standard machine learning problem consists of building a model that takes processing conditions as input and produces a prediction of material properties as output, the scientists we are working with wanted to understand all those processing conditions that would give a fixed material property (they could then sort through these and find those that worked well with various experimental set-ups). This was our motivation for sampling from the fiber of a task. Unfortunately, we found that describing the data and technical details involved in the science component of this problem was non-trivial and worried that inclusion of this application in the paper would be a confusing distraction. Your suggestion of at least mentioning this motivation makes sense however, and we have added a sentence to the end of the second paragraph in the introduction.
> - We also expect that scaling our method up to higher-dimensional datasets will likely require additional adaptation. This will be our central goal for follow-on work. As the reviewer suggested, we view the present work as (1) outlining the problem of sampling from a fiber and (2) presenting one topologically inspired approach that empirically works well in low dimensions. Our restriction to low-dimensional data was also related to the low-dimensionality of the material science data that was described above.
>
> Again, we thank the reviewer for their helpful comments which undoubtedly have made the paper stronger.

---

> > ### Comment · Reviewer_asbk · 2021-11-29
> > **Score unchanged**
> >
> > I thank the authors for their added section and comments. After reading the other reviews I feel that perhaps I was a little hasty in my initial assessment. But I still think that a 6 would be too weak a score and will keep my recommendation fixed at 8.

---

### Official Review · Reviewer_6GTJ · 2021-11-04

**Correctness:** 4
**Technical Novelty And Significance:** 2
**Empirical Novelty And Significance:** 3
**Recommendation:** 6
**Confidence:** 4

**Main Review:**

### Strengths of the paper

- I like the idea to model many-to-one tasks in machine learning using fiber bundles. I find this natural and elegant. The authors do a good job of motivating introducing and motivating the model.
- The proposed architecture is natural and more or less directly maps the fiber bundle technology to neural networks to learn fiber bundles from data.
- The author propose an effective training strategy to optimize the parameters of the proposed model which includes multiple well-motivated loss functions.
- Empirically the proposed architecture seems to outperform conditional generative models.
- I very much appreciate the discussion of the topological aspects and the exploration of the influence of topology of the latent space on performance.
- Overall the paper is well written and easy to follow. The illustrations help understand the technical points in the text.


### Weaknesses of the paper

- At a high level the fiber bundle idea (in a machine learning context) is very similar to that of conditional generative models. The authors argue that a specialized architecture is warranted and that networks like conditional flows cannot work well. These discussions are however at a rather heuristic level, with argument such as ... . To me it remains unclear why conditional networks underperform by such a significant margin.
- One claim the authors make is that the improved performance of their model (relative to conditional generators) comes from space partitioning, but partitioning can be interpreted as a particular choice of conditional architecture. In conditional generative models the only conditioning object is $y$; same seems to hold in BundleNet except that $y$ enters the architecture in two different ways, once through the argmin calculation of the nearest mean, and once on the "reverse" input on the right hand side. The question that comes to mind is then: why not use a slightly more advanced conditional generator architecture? My gut feeling is that it should perform well.
- I am uncertain about the significance of the "CGAN-nbhd" model in your comparisons. You state that this model is your best effort at modifying a standard architecture to sample from fibers, but its performance is _worse_ than that of its unmodified version. I am not sure about the message: that including fiber-bundle-specific details in a model makes its performance worse? Arguing from empirics, it means that the unmodified model is closer to the bundle ideal, no? I would suggest to remove this model from all comparisons and perhaps add a proper "in-between" model that actually does better than vanilla conditional generators. It is impossible to interpret why the performance of CGAN-nbhd is the way it is (it may have to do with things like hyperparameters, architecture details, training strategy, ...)
- Overall, I would like to see a much more thorough and convincing comparison between the various models and a better effort at training the existing conditional architectures. (Again, there are no convincing hypotheses, tested numerically, for why conditional generative models perform worse.)
- The current architecture seems to be suitable for very low-dimensional datasets; things like $k$-means are hard to motivate in higher dimension. Could you discuss examples and implications of applying your ideas to high-dimensional problems?
- I believe that considerable care is required in interpreting the empirical statements about the influence of topology. I am not convinced that these empirical results are broadly representative (they are likely to strongly depend on the chosen models and hyperparameters). What is more, many real-world data distributions are believed or known to be low-dimensional yet invertible flows model them without difficulty. The example shown by the authors might be an artifact of low dimension for two reasons: (topological) low-dimensional topology and embeddings are more quirky than high-dimensional and (expressivity) the flow networks used have limited expressivity when approximating functions $\mathbb{R}^2 \to \mathbb{R}^2$. They might work better in high dimension.
- A minor point: I would appreciate having the dataset descriptions in the main text. The main results of the paper are empirical, on Airfoil Noise and Wine Quality datasets—it is hard to build intuition without knowing the relevant details of the datasets. One would need to go to the appendix, but even there you refer to some external resources.

### Notes and questions
- Out of curiosity: your explore the case where the fiber bundle structure is essentially known. Are there applications where such a structure would have to be discovered? Could you adapt your network to these scenarios?
- Is there a connection between your model and the recently-proposed multichart flows? (I am thinking about the label space partitioning)
- How are the conditioning vectors $r_i$ structured? (are they simply scalars or they have the structure of labels $y$ or inputs $x$?)
- Since you're using an invertible network, it seems to me that $\text{dim}(y) + \text{dim}(z) = \text{dim}(x)$, but $x$ corresponds to points in the ambient space. Many datasets are believed to be intrinsically low-dimensional. Would it then make sense to use injective flows like https://arxiv.org/abs/2102.10461 or https://arxiv.org/abs/2003.13913 instead of invertible flows?


**Summary Of The Paper:**

This paper proposes a neural network to model data (or machine learning tasks) with fiber bundle structure. Machine learning tasks like regression or classification are often many to one. They assign a "label" (discrete or continuous) $Y \ni y = \pi(x)$ to each $x \in X$. Many different $x$ may receive the same label. It is interesting to explore / parameterize the set of all $x \in X$ that yield a given label $y$, that is to say, the set $\pi^{-1}(y)$. This is nicely modeled by the topological construction of a fiber bundle (a space that is locally a product space). Operationally, it corresponds to a covering of the label space $Y = \bigcup_{i} U_i$ by "patches" $U_i$ so that $\pi^{-1}(U_i)$  are homeomorphic to product spaces $U_i \times Z$, with $Z$ being the fiber space.

The proposed network is in essence a conditional generative model with the added twist that the conditioning is also performed on the "patch" that a label (or a regression target) belongs to (for training; inference is only performed in the "reverse" direction of fiber sampling.)

The authors show interesting results in modeling familiar low-dimensional bundles as well as real-world datasets. Comparisons with conditional generative models suggest that the new architecture may be a better fiber bundle model. Finally, the authors numerically explore the role of "latent space" topology, a welcome addition.


**Summary Of The Review:**

This interesting paper suggests that fiber bundles are good models of many-to-one tasks in machine learning. The authors propose a deep neural network to model fiber bundles and argue that it outperforms conditional generative models. I find the premise of the paper natural and convincing, but the numerical results, evaluations of conditional generative models, and the empirical claims about the influence of topology lacking, hence my recommendation. Overall I think the paper has a lot of potential but the narrative and the comparisons should be much more solid, especially when it comes to existing conditional generative models and diagnosing the (topological) modes of failure. My initial recommendation is thus a rejection but I am more than willing to adjust it after the author response. (I would've chosen a 4 but it is not available anymore.)

---

> ### Author Response · Authors · 2021-11-19
> **Response**
>
> We would like to start by thanking the reviewer for a thoughtful and very helpful review. We have little doubt that implementing the changes that were suggested have made the paper much stronger overall. We believe that those points that we were not able to comprehensively address will be an important component of any future work.
>
> Below we go through those weaknesses in the paper that the reviewer identified and comment on each, including additions we made to try and address them.
>
> - Better training of conditional models: The reviewer rightly pointed out that we did not include much information about why the CGAN model that we benchmarked against did not perform better on the datasets we used. The reviewer also observed that including CGAN-nhbd did not make much sense if it could neither compete with BundleNet nor CGAN. We tried to address both these by revisiting our experiments using the CGAN and CGAN-nbhd models. Various exploratory experiments made us confident that CGAN-nbhd should be able to exceed the performance of CGAN. As explained above, after experimenting with both models we found that by changing the batching procedure that we were using when training CGAN-local (our new name for CGAN-nbhd) we could dramatically improve performance, outpacing the performance of the vanilla CGAN. While there might also be a trick for improving the performance of the vanilla CGAN, we were not able to find it in our hyperparameter exploration. Being that CGAN-local can be viewed as an interpolation between CGAN and BundleNet, we interpret its intermediate performance between CGAN and BundleNet as suggesting that while providing neighborhood information to a CGAN can improve performance, the specialized invertible architecture and loss function combination of BundleNet makes it a better model for the fiber reconstruction task. This is now documented in the paper in Section A.4.1 (as well as summarized in Section 5.2 in the main body of the paper).
> - Higher dimensional data: As mentioned above, we strongly agree with the reviewer that testing (and likely adapting) our approach to high dimensional data is the most important next step. As we mention now in the introduction, our inspiration for this paper comes from a problem in materials science involving (relatively) low-dimensional data, so this was the setting where we started. A follow-on work generalizing to high-dimensional data would doubtless increase our impact.
> - More extensive comparison among models: The reviewer mentioned that they would have liked to see more extensive comparison between the different models that we tested. We tried to add some content addressing this in Section A.4. Our main takeaway, at least in the synthetic datasets, was that WGAN (as would be expected) and CGAN (more surprisingly) struggle to reconstruct individual fibers. CGAN-local performs much better, but even it suffers from exaggerated clumping of neighborhoods and smearing at the twist of the Moebius band and singularity of the sliced torus.
> - Topological statements: To try to address the reviewer’s concerns that we draw conclusions from limited synthetic experiments, we added a caveat to the end of Section 6. We also include some data in section A.3.2 on the effect of different topological choices with respect to one of our real-world datasets. In contrast to the synthetic dataset, this experiment showed that BundleNet’s performance is relatively robust to different choices of circle and Gaussian prior. We have updated the statements in the paper to reflect these new results.
> - Dataset descriptions: To give more clarity regarding the real-world datasets that we used, we followed the reviewer’s advice and gave a longer description of both Airfoil Noise and Wine Quality in the body of the paper (Section 5).
>
>
> (continued in next comment)

---

> > ### Author Response · Authors · 2021-11-19
> > **Response (cont.)**
> >
> > Finally, we address some of the reviewer’s notes and questions.
> >
> > - Form of r_i’s used in conditioning: Experimentally we found that BundleNet worked best when each r_i was actually the centroid extracted by k-means. Thus r_i was a vector from Y. Details for this can be found in the second paragraph of Section 4.
> > - MCF paper: We thank the reviewer for pointing out the work of Kalatzis, et. al., on multi-chart flows. The reviewer is correct that this work also leverages the idea of partitioning a space into neighborhoods. As we understand it, Kalatzis does this to enable the use of Euclidean machinery to model topologically nontrivial manifolds. This is slightly different than our work, where we partition input space into neighborhoods under the assumption that it will be easier to learn maps that decompose variation related to label and variation not related to label locally. The difference amounts to partitioning a space to capture local Euclidean-ness vs local trivialization. We added a short paragraph to our Related Works covering this connection.
> > - Injective flows: Use of injective flows seems like a promising strategy for dealing with issues that arise when BundleNet is used on data living in a high dimensional ambient space (such as natural imagery for example). It does seem likely that in certain cases it would be more natural to have dim(Y) + dim(Z) < dim(X). We thank the reviewer for pointing this tool out as it seems likely to be a useful model for our next round of research.
> > - Other fiber bundle structures: We assume that the reviewer means fiber bundle structures other than those defined by the labels attached to data. We think that this is a great question. In fact, it may be that for other modes of variation a fiber bundle structure would be more likely to reflect the data itself. As another reviewer pointed out, for some problems it is unlikely that the label space defines a fiber bundle “on the nose”. Given existing results in feature disentanglement, we suspect that applying a fiber bundle structure to other aspects of a dataset may in some cases be even more natural.
> >
> > We again thank the reviewer for all their useful feedback.

---

> > > ### Comment · Reviewer_6GTJ · 2021-11-23
> > > **A great update**
> > >
> > > I'd like to thank the authors for a substantial revision which addresses many of my comments. While some things could still be improved (a convincing explanation for the poor performance of conditional models, what happens if we try this on high-dim data (conditional generative models work with images), necessity that the bundle be full-dimensional, and what about theoretical results?), I am increasing my score to 6. I should say: I would've liked to choose a 7 but it's not available!

---

> > > > ### Comment · Reviewer_6GTJ · 2021-11-23
> > > > **Still thinking...**
> > > >
> > > > Let me also add: the above remaining drawbacks are preventing me from giving an 8 (I thought about this for a while but a 6 seemed more appropriate.) That said, I will wait to read other reviewers' post-revision updates and keep thinking about it.

---

### Author Response · Authors · 2021-11-19
**Summary of changes and responses**

We would like to start off by thanking all the reviewers for taking the time to carefully read through our paper. There is a lot of great feedback in each of the reviews. Where time has permitted it, we have tried to address weaknesses in the paper that were pointed out by reviewers. In other cases, we plan to use feedback as the basis for follow on work. In our response to each review below we have outlined how we changed the paper based on the feedback we received. Here we give some general remarks about changes to the paper. We include more detailed responses when we address individual reviewers:

- Improved and renamed CGAN-nhbd: Several of the reviewers commented on the inclusion of CGAN-nbhd. CGAN-nbhd was meant to be a model that used the exact same information as BundleNet (in particular, neighborhood representatives for conditioning) while maintaining a GAN framework. Unfortunately, CGAN-nbhd achieved worse performance than both CGAN and BundleNet. Following reviewer feedback we did a study of hyperparameters and training routines. We realized that the failure of CGAN-nbhd could be attributed to the batching strategy we were using (specifically, we switched from creating batches with instances from many neighborhoods to batches with instances from a single neighborhood). The improvement of CGAN-nbhd performance made it second only to BundleNet among all the models we studied. **We also changed this model’s name from CGAN-nbhd to CGAN-local**, as we felt the new name more clearly communicated the difference between this model and a vanilla CGAN. We added Section A.4.1 that describes the importance of training routine for the CGAN-local. This section includes results from the old method of training.
- Revisiting all experiments: In line with some of the changes described in the previous bullet point, we re-ran all our experiments and updated our tables. BundleNet still outperforms other models on almost all datasets (with the exception of Airfoil Noise where CGAN-local ties with BundleNet on a few of the fiberwise metrics).
- Hyperparameter study: Following comments from several of the reviewers, we included two new sections devoted to hyperparameters associated with BundleNet. This includes Section A.3.1 (which looks at robustness to the number of neighborhoods used) and A.3.2 (which looks at how topological priors affect BundleNet performance).
- More extensive model comparison: To give a more thorough analysis of model performance for different models, we added Section A.4 which provides visualizations of synthetic dataset reconstruction for each of the models. We hope that this will begin to illuminate some of the failure modes of each of the model types.

Finally, several reviewers pointed out that our work does not address issues that would likely be encountered when moving from low-dimensional to high-dimensional data. Our motivation for this paper arose from a problem in materials science wherein one wants to understand all processing conditions that will lead to a fixed set of properties in a material (this motivation has now been added to the Introduction based on some of the feedback we received). The data for this problem is relatively low-dimensional, which is why we first tackled this problem in the low-dimensional regime. However, we strongly agree that understanding how to adapt it to high-dimensional data should be the top next priority. We unfortunately were not able to explore this aspect of the problem yet, but we plan to make this the central effort of future work.

---

### Decision · Program_Chairs · 2022-01-20

**Decision:**

Accept (Poster)

**Comment:**

The paper studies the problem of learning fiber distributions associated with a machine learning task, in which the goal is to predict Y, given X. One chooses a fiber space / distribution Z / D_Z, and learns a trivialization \varphi : (Y,Z) -> X. The proposed architecture first clusters the label space Y. Within each cluster i it fixes a fiber space and distribution, and then learns a mapping \Phi_i, parameterized by an invertible neural network, by minimizing the discrepancy between the generated distribution and the distribution of the training data. The paper performs experiments on the wine dataset and a dataset coming from an aerospace application, as well as synthetic data with fiber bundle structure. Since the task here is generative modeling, the paper compares to standard GAN architectures (WGAN and conditional GAN) and argues that they are not fiber-learners, in the sense of this paper.

Initial reviews were split, with reviewers appreciating the novelty of the fiber learning task, while also raising questions about the paper’s relationship to conditional GANs, some points of clarity, and limitations of the experiments. After interaction in the response period, the reviewers converged to a decision to accept. The paper’s primary strength is its clear formulation — the paper provides useful language for describing conditional generative models (in particular, for discussing when a factorization of the distribution over Y and Z is appropriate), a valuable contribution to the discussion in this area.